# Comparing fNIRS signal qualities between approaches with and without short channels

**Xin Zhou** [1]*, **Gabriel Sobczak**[2], **Colette M. McKay**[3,4], **Ruth Y. Litovsky**[1,5,6]

**1** Waisman Center, University of Wisconsin-Madison, Madison, Wisconsin, United States of America,
**2** School of Medicine and Public Health, University of Wisconsin-Madison, Madison, Wisconsin, United States of America, **3** Bionics Institute of Australia, Melbourne, Australia, **4** Department of Medical Bionics, University of Melbourne, Melbourne, Australia, **5** Department of Communication Science and Disorders, University of Wisconsin-Madison, Madison, Wisconsin, United States of America, **6** Department of Surgery, Division of Otolaryngology, University of Wisconsin-Madison, Madison, Wisconsin, United States of America

* xzhou353@wisc.edu

**Data Availability Statement:** All relevant data are within the manuscript and its Supporting information files.

**Funding:** The work was supported by NIH-NIDCD (https://www.nidcd.nih.gov/, R01DC003083 to

## Abstract

Functional near-infrared spectroscopy (fNIRS) is a non-invasive technique used to measure changes in oxygenated (HbO) and deoxygenated (HbR) hemoglobin, related to neuronal activity. fNIRS signals are contaminated by the systemic responses in the extracerebral tissue (superficial layer) of the head, as fNIRS uses a back-reflection measurement. Using shorter channels that are only sensitive to responses in the extracerebral tissue but not in the deeper layers where target neuronal activity occurs has been a 'gold standard' to reduce the systemic responses in the fNIRS data from adults. When shorter channels are not available or feasible for implementation, an alternative, i.e., anti-correlation (Anti-Corr) method has been adopted. To date, there has not been a study that directly assesses the outcomes from the two approaches. In this study, we compared the Anti-Corr method with the 'gold standard' in reducing systemic responses to improve fNIRS neural signal qualities. We used eight short channels (8-mm) in a group of adults, and conducted a principal component analysis (PCA) to extract two components that contributed the most to responses in the 8 short channels, which were assumed to contain the global components in the extracerebral tissue. We then used a general linear model (GLM), *with* and *without* including event-related regressors, to regress out the 2 principal components from regular fNIRS channels (30 mm), i.e., two GLM-PCA methods. Our results found that, the two GLM-PCA methods showed similar performance, both GLM-PCA methods and the Anti-Corr method improved fNIRS signal qualities, and the two GLM-PCA methods had better performance than the Anti-Corr method.

## Introduction

Functional near-infrared spectroscopy (fNIRS) is a non-invasive technique that uses near-infrared light to measure the concentration changes in the oxygenated (HbO) and deoxygenated (HbR) hemoglobin in brain tissue. Since changes in hemoglobin in the cerebral blood flow are associated with the changes in local neural activity through neurovascular coupling

RL), UW-Madison's Office of the Vice Chancellor for Research, and a Core grant from NIH-NICHD (U54HD090256 to Waisman Center). The sponsors and funders were not involved in the study design, data collection and analysis, decision to publish, or preparation of the manuscript.

**Competing interests:** Dr. Litovsky discloses that she is a consultant for Frequency Therapeutics; however, this does not alter our adherence to PLOS ONE policies on sharing data and materials. Other authors have no conflicts of interest to disclose.

[1], fNIRS has been implemented in human brain imaging research [see reviews by 2,3]. Currently, fNIRS faces a major challenge due to confounding signals in the data causing poor signal-to-noise ratios (SNRs). The confounding signals include motion artifacts [4], systemic activity such as respiration, heart rate, low-frequency fluctuations in blood pressure [5,6] and Mayer waves [7] in both the extracerebral and cerebral tissue [8], and spontaneous (non-evoked) neural activity in the cerebral tissue [9]. The current study compared two approaches that have been widely used to improve fNIRS signal quality by reducing systemic activity and noise from extracerebral tissue.

fNIRS uses a back-reflection measurement from a light source to an associated detector. As near-infrared light travels in and out of the extracerebral tissue, it passes through highly vascularized regions, including the scalp and skull, which have an average total thickness of about 13 mm [10]. fNIRS is very sensitive to the responses in extracerebral tissues [11] and physiological signals and comparatively less sensitive to changes in hemoglobin in the deeper cerebral tissue [12]. Therefore, removing the extracerebral components and physiological signals in the fNIRS signals is crucial for improving the fidelity of measures of neuronal activity. Several methods have been used in the literature to reduce the systemic activity in fNIRS signals. The first method [13] applies a low-pass filter, e.g., with a cut-off frequency of 0.1 Hz, to remove respiration and cardiac signals. However, this method fails to remove the low-frequency (below 0.1 Hz) cardiovascular oscillations in the blood flow such as Mayer waves that overlap in the frequency domain with neuronal responses recorded using block design [7,14].

A second method [15–18] performs a principal component analysis (PCA) on the fNIRS signals to remove motion artifacts and extracerebral responses. This method assumes that noise sources including extracerebral responses and motion artifacts that are much larger in amplitude compared to physiological signals, are homogenous across channels thus contribute the most to signals in all the channels. In this way, regressing out the first few principal components (PCs) in fNIRS signals would reduce the motion artifacts and the global component in the extracerebral tissue. This method is also referred to as a spatial filter. Another method that is also based on spatial separation, independent component analysis (ICA), has been used in some studies to reduce systemic noise without short channels [19–21]. The study reported in [21] compared the efficiency of PCA and ICA methods in reducing the systemic responses in fNIRS signals. Their results found that both methods were able to reduce the systemic responses, but the ICA method was inferior to the PCA method. However, when motion artifacts are rare in the fNIRS data, the assumption that motion artifacts are present in the first few PCA components would no longer be valid and instead lead to a risk of reducing the wanted neuronal signals in the fNIRS data. Further, the spatial separation methods assume that the extracerebral responses are not correlated with or independent from neuronal signals in the cerebral tissue, which might not be valid. In the cases when the extracerebral responses and cerebral responses are correlated, the PCA and ICA methods can not separate them to further remove the extracerebral responses in the fNIRS signals.

Besides the spatial separation methods (PCA and ICA) as explained above, an anti-correlation method (Anti-Corr) was proposed [22,23]. This method assumes that the changes in neuronal HbO and HbR responses are anti-correlated, whereas the systemic-activity or noise-related changes in HbO and HbR are not [23]. For example, motion artifacts cause positive correlations between changes in the HbO and HbR responses [22]. Due to a relatively high efficacy in reducing the systemic noise in fNIRS signals, the Anti-Corr method has been widely used to reduce systemic activity in fNIRS signals [24–27]. However, the fundamental assumption of the Anti-Corr method may not be entirely valid, as changes in neuronal HbO and HbR responses are not perfectly anti-correlated, with HbR changes lagging in time by ~1-2s or so

behind HbO changes [28]. Hence, the anti-correlation method is likely to yield errors when used to quantify changes in neuronal activity.

Inspired by the PCA method, a more refined method known as the short-channel subtraction method has been introduced to capture and reduce systemic responses in the fNIRS signals due to extracerebral tissue activity [29–31]. The short-channel subtraction method is based on two assumptions. First, the short channels are configured on the scalp such that only responses from extracerebral tissue are measured, without contribution from the relatively deeper cerebral tissue [32]. Secondly, systemic responses such as respiration, heartbeat, Mayer waves, changes in blood pressure and cutaneous blood flow in the extracerebral tissue are partially homogeneous. Therefore, the global component measured by the short channels also exists in signals from the regular fNIRS channels [33]. Using a general linear model (GLM) regression analysis a GLM-PCA method can be applied to remove the short-channel component from the regular fNIRS channels [30,31,34]. The GLM-PCA method with short channels compared to PCA and ICA methods with no short channels mentioned above are not suited to handle the issue of failing to separate the extracerebral and neuronal responses, if they are correlated. Besides the GLM-PCA method, some studies used a linear regression [32,35] or an adaptive filtering [36,37] to subtract responses recorded using shorter channels from signals in the regular fNIRS channels. The linear regression and adaptive filtering assume that the local scalp responses measured by short channels also exist in the paired regular fNIRS channel. Hence, these 'direct' subtraction methods were often adopted when dual optodes were used, i.e., with a light source connected with one detector at closer (less than 10 mm) and one at regular distances (30 mm). The direct subtraction method was also adopted when short channels were within 15-mm distance from regular channels, center-to-center [34]. However, in studies that use many regular channels, pairing each regular fNIRS channel with one or more shorter channels is technically impractical and costly.

The short-channel subtraction has been considered as the 'gold standard' to reduce the responses from the extracerebral tissue in fNIRS data. However, there does not appear to be agreement in the literature on the ideal range for source-detector distances. Brigadoi and Cooper [38] used Monte Carlo simulations to propose that source-detector distances for adults should be no longer than 8.4 mm to optimize extraction of responses from extracerebral tissue without the inclusion of cerebral tissue. However, published studies have used short channels with distances of 13–15 mm [30,31,33,34,39,40], 20 mm [41] and even up to 25 mm [42]. The potential difficulty with using larger source-detector distances for "short channels" is that statistical regression to data from the regular fNIRS channels can introduce false negatives into the results. That is, by regressing out the short-channel components, which possibly included cerebral responses, the cerebral responses in the regular fNIRS channels would also be reduced. The study reported in [43] found that source-detector distances of 13-mm yielded poorer outcomes in improving fNIRS signal qualities compared to 6-mm distances.

Thus far, we have argued that shorter source-detector distances are generally better for lowering false-negative rates, and channels as small as ~5 mm have been introduced [32]. However, very short distances between light sources and detectors can also introduce challenges, because with shorter distances, the detectors can be saturated with photons [38]. To date, manufactures have been able to produce 8-mm short channels [7] that seem to successfully mitigate light saturation problems and minimize the amount of photons in short channel photon paths entering into cerebral tissue layers.

Despite the current 'gold standard' of including short channel in fNIRS studies, there is a paucity of research on the functioning of short channels, including data quality at the individual level and responses that have been measured. Hence, the current study provided detailed information on the functioning of eight 8-mm short channels for references when future

studies consider whether to include short channels. We employed the GLM-PCA method by including the first two principal components (PCs) from the short channels to represent the event-related and non-related systemic response in the extracerebral tissue. Two GLM-PCA methods were conducted, with and without including stimulus-related regressors, to identify whether the event-related systemic response dominated in the extracerebral tissue. We further compared the efficacy of reducing systemic responses hence improving fNIRS signal quality between using the GLM-PCA methods and the Anti-Corr method without short channels to evaluate the necessity of including short channels. To evaluate fNIRS measures of neuronal activity evoked by an auditory task versus noise, we calculated the contrast-to-noise ratios (CNRs) from the HbO responses, which has been widely used to assess fNIRS signal quality [18,20,44]. We expected that the GLM-PCA methods with short channels would perform better at improving fNIRS signal quality, i.e., showing greater CNR values compared to the Anti-Corr method.

## Methods

### Participants

Twenty-three adults (10 males, mean and standard deviation (SD) of age: 22.70 ± 3.07 years) participated. This study was approved by the University of Wisconsin–Madison's Human Subjects Institutional Review board and all the participants gave written consent. All participants were native English speakers, students at the University of Wisconsin-Madison. All participants passed the pure-tone audiometry using air conduction, with hearing thresholds less than 20 dB HL at octave frequencies between 125 Hz and 8000 Hz.

### fNIRS data collection and short channels

The current study used a continuous-wave NIRS instrument (NIRScout system, NIRx medical technologies, LLC) with 16 LED light sources (Fig 1A, red arrows) and 16 avalanche photodiode (APD) detectors (Fig 1A, blue arrows) for data collection, with a sampling frequency of 3.91 Hz. Each LED light source emitted near-infrared light with wavelengths of 760 nm and 850 nm. A light source paired with detectors located at about 30-mm distance provided fNIRS channels that collected signals. In total, forty-four 30-mm (regular) channels were included and grouped into twelve regions of interest (ROIs) based on their anatomical positions, with ROI1–ROI6 on the left and ROI7–ROI12 on the right side (Fig 1C). These ROIs covered the ventral-lateral pathways of cortical regions for auditory speech processing, including the frontal cortex, inferior prefrontal, anterior temporal cortex, and the superior temporal areas (see reviews with meta-analyses in [45,46]). As shown in Fig 1C, each ROI consisted of 4 adjacent or crossing channels (labeled circles), which were aimed to measure signals from neighboring local brain regions. Except that ROI2 and ROI6 only consisted of two crossing channels. A NIRScap (NIRX medical technologies, LLC) matched to each participant's head circumference was used to hold the light sources and detectors on the head. The optodes were located based on the standardized 10–10 system [47]. As the distances between some source-detector pairs on the NIRScap were above the optimal distance (30 mm) for fNIRS data recording, plastic spacers were used to reposition the two optodes and keep the distances between them at 30 mm.

Eight 8-mm channels were also used, whereby an extra silicon photodiode detector was split into 8 groups of dual-detectors (Fig 1A, green arrows), with each group surrounding a light source at 8-mm distance (Fig 1C, green dots with green lines). Panel (B) shows a schematic diagram of the connection between a light source and a short-channel detector, or

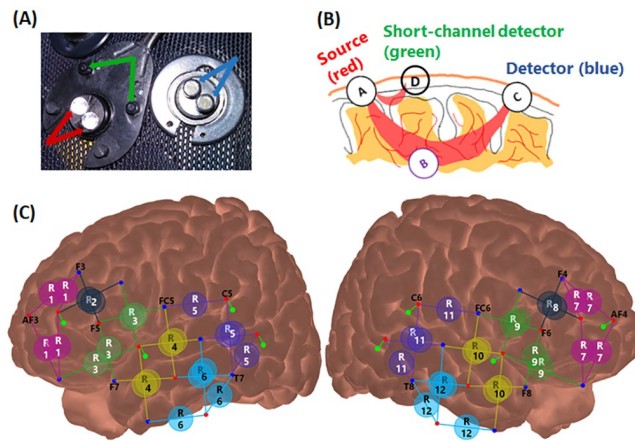

**Fig 1. Short-channel setting and fNIRS montage for data collection.** Panel (A) shows the light source and paired short-channel or regular detectors. Panel (B) shows a schematic diagram of light sources and detectors connections and light travelling through the brain tissue. Pairing a light source (in red) and a regular detector (in blue) at 30 mm distance provides a regular fNIRS channel, which measures response from the local region between them, including from the shallower extracerebral tissue and deeper cerebral tissue. Pairing a short channel detector (in green) at 8 mm distance from the light source provides a short channel, which measures signals only from the extracerebral tissue. Panel (C) shows the montage of optodes for fNIRS data collection. Red dots and blue dots are for light sources and regular detectors, respectively. The lines between them and labeled circles indicate the clusters of channels into regions of interest (ROIs). Green dots with green lines connected to the light sources (red) are for the short-channel detectors. Panel C was generated with software NIRSite and then adapted.

between a light source and a regular detector. The eight short-channel detectors were located on two hemispheres, with 2 on the inferior frontal and 2 on the temporal lobes of each side.

The study was conducted in a standard acoustics double-walled sound booth designed by the company Industrial Acoustics Company (IAC), while participants sat in an armchair wearing a NIRSCap of suitable size. During preparation, to center the cap and correctly position the optodes on the head, the Cz was positioned halfway the distance of Nasion to Inion and halfway the distance between the two pre-auricular points. Further, the frontal electrodes Fp1 was positioned at 10% of the Nasion-Inion distance (a few centimeters above the eyebrows). The cap was attached to a chest wrap for fixation. Then the gains of light intensity at the APD detectors were checked to ensure that all the channels had at least 'acceptable' light intensity. In the event that a channel did not show good intensity, poor contact between the optodes with the skin usually accounted for this error, i.e., optodes in a poor angle and hair artifacts. To correct this issue, optodes were first removed from the NIRSCap to push away the hair from the contact zone and ensure perpendicular contact with the scalp, and then replaced. The procedure was repeated until most of the optodes receive at least 'acceptable' light intensity shown on the data collection software interface.

## Stimuli and experimental design

Auditory speech stimuli consisted of sentences with a uniform structure (name, verb, number, adjective, and noun), for instance, '*Bob has six green hats*'. Sentences, unprocessed or with a noise-vocoder [48], were presented to the left ear, in the presence of background four-talker speech babble that was presented to the left ear or both ears simultaneously. The babble was also filtered with a noise-vocoder, and presented at a speech-to-noise ratio (StNR) of -10 dB or -15 dB. In total, eight hearing conditions were involved during data collection.

fNIRS data were collected using a pseudo-random block design in three separate sessions. In each session, stimuli in four different conditions at the same StNR (-10 dB or -15 dB) were

presented in a random order. Each session started with a 30-s silent baseline; after each block of 13.6-s stimulation consisting of 5 sentences, there was a silent period ranging from 25 to 35 s. Nine blocks of stimuli per condition were presented in total. Throughout the test, there was a white cross on a black monitor 1.5 m in front of the participants. Participants were given verbal and written instructions to look at the cross and avoid dramatic head movement during data collection. They were also required to attend to the speech sentences and count the number of color words in each block and to push mouse buttons to indicate an even or odd number of colors at the end of each block. fNIRS data reported in this study, was part of a project that examined listening effort in normal hearing listeners under a total of 8 hearing conditions. As the aim of the current study was to examine the efficacy of the short-channel subtraction method but not about listening effort, we presented fNIRS data in one of easy hearing conditions only, i.e., unprocessed speech presented with noise in both ears at -10 dB StNR. The experiment was run in the neurobehavioral Presentation software (https://www.neurobs.com/).

## fNIRS data analysis

The fNIRS signals recorded by the NIRScout system were exported to MATLAB (MATLAB R2017a) for further analysis, with a combination of customized software written by the authors and scripts adopted from Homer2 [13]. The steps taken in the signal processing are listed below and are also illustrated in Fig 2.

1. **Remove step-like noise**. Step-like noise can be caused by sudden loss of contact between optodes and the skin, or interposition of hair, during data collection. If not removed, the step-like noise would skew fNIRS signals. To remove step-like artifacts in the data of each channel ($y$), the derivative of y was first estimated as $X$. Any absolute values in $X$, i.e., abs($X_i$), which were two standard deviations or more above the mean of the absolute values of $X$, were set as zeros. Response $y$ (with step-like artifacts removed) was then recovered by calculating the cumulative sum of the updated $X$.

2. **Exclude channels of poor quality**. A third-order Butterworth bandpass filter with cut-off frequencies of 0.5 and 1.5 Hz was applied to raw fNIRS intensity data to extract heartbeat signals in individual channels. Channels that showed poor correlations of heartbeat signals between the two different NIR wavelengths, which has been called the scalp coupling index (SCI) in Pollonini et al. [42], were excluded from further analysis. Because the heartbeat is a characteristic component of the fNIRS signal, channels that fail to record the heartbeat signals, i.e., poor SCI, are unlikely to record other physiological or neural responses. In the study by Pollonini et al. [42], channels with SCI less than 0.75 were excluded. In the current study, the 8-mm short channels had relatively poor success in quite a few subjects (Fig 3A) likely due to hair artifacts. Hence, a lower cut-off frequency (SCI > = 0.15) was set compared to that in [35] to ensure for each participant there were at least 4 shorter channels remaining for further analysis of the data from each session. The number '4' was decided as Sato et al. [31] reported that a GLM-PCA method with 4 short channels provided a robust estimation of cerebral activity. Further, keeping the threshold of SCI > = 0.15 ensured that short channels from both the frontal cortex and the temporal cortex were included in the PCA, to better represent the common components in the extracerebral tissue across a wide brain region.

3. **Convert light intensity to optical density**. See Huppert et al. [13].

4. **Correct motion artifacts**. The motion artifacts might be caused by the physical displacement of the optodes from the surface of the participant's head. The wavelet decomposition

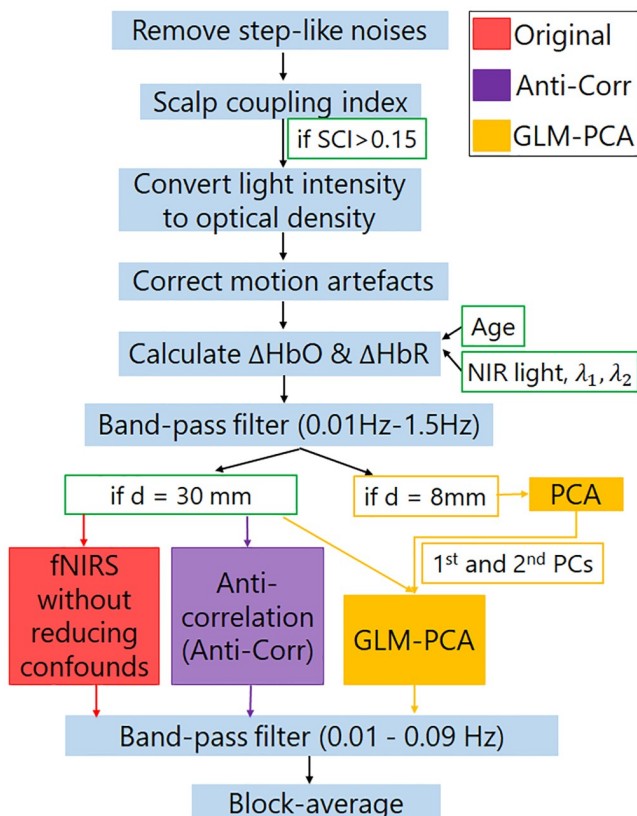

**Fig 2. Diagram of fNIRS signal processing and systemic response reduction.** SCI is for scalp coupling index. 'd' is for the distance between source and detector in an fNIRS channel.

method proposed by Molavi and Dumont [49] was then performed to correct motion artifacts. With wavelet decomposition, motion artifacts appear as abrupt breaks in the wavelet domain, whereas hemodynamic responses to stimuli have less variable coefficients. To remove the motion artifacts, wavelet coefficients out of the interquartile range of 0.1 were set to zero, the same setting as in Zhou et al. [25].

5. **Calculate the concentration changes of HbO and HbR**. To adjust the effect of age and wavelengths of near-infrared light on the calculation of ΔHbO and ΔHbR, the method proposed by Scholkmann and Wolf [50] was used to calculate wavelength-dependent differential pathlength factor. The optical density data were then converted into the ΔHbO and ΔHbR using the modified Beer-Lambert law [51].

6. **Calculate the block-average of ΔHbO and ΔHbR without or with reducing systemic responses**. For all the 8-mm and 30-mm channels with SCI above 0.15, block-average fNIRS responses were first calculated without using the GLM-PCA or the Anti-Corr method. A third-order Butterworth band-pass filter (cut-off frequency at 0.01–0.09 Hz) was applied to remove high-frequency physiological signals, such as respiration and heartbeats (Fig 2, red box). Block-average fNIRS responses were then calculated, with baseline averages of each block, i.e., 5 s before stimulus onset, being subtracted. Responses in each ROI of individual participants were calculated as the grand mean of block-average fNIRS responses across channels clustered in the ROI.

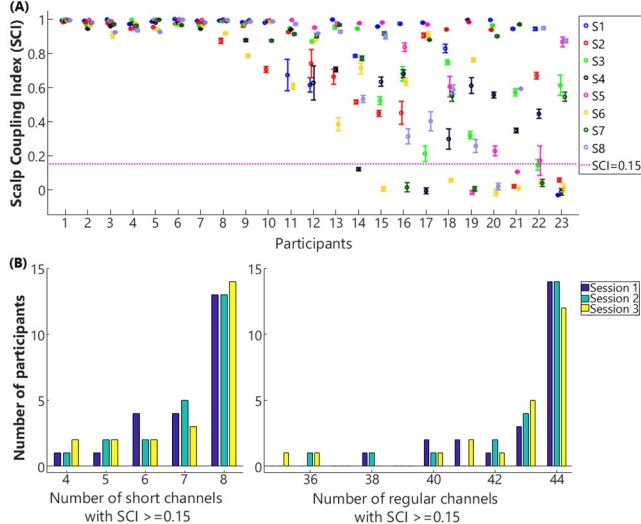

**Fig 3. Scalp coupling index (SCI) in the short channels and the number of channels involved for data analysis.**
Panel A plots the mean (circles) and standard deviation (bars) of SCI values across three sessions for the 8 short channels, i.e., S1—S8. The magenta horizontal dash line denotes the cut-off threshold (SCI = 0.15); channels that showed SCI less than the threshold were excluded from further analysis. Panel B shows the histogram of numbers of regular (right) and short channels (left) that were included in the GLM-PCA methods for each session of data collection among participants.

7. **Reduce systemic responses**. The Anti-Corr method proposed by Cui et al. [22] was used to reduce the systemic responses (Fig 2, purple box) without short channels. The Anti-Corr method was applied to the calculated concentration changes of HbO and HbR for the regular channels (step 5). Then the same band-pass filter and block-average as explained in step 6) were conducted.

8. **Reduce extracerebral components**. The GLM-PCA method was applied to reduce the extracerebral components in the HbO and HbR responses, separately (Fig 2, yellow box) in the regular channels. For fNIRS signals in both short and regular channels, a third-order Butterworth band-pass filter (0.01–1.5 Hz) was applied to remove the low- and high-frequency physical noise. A PCA was then performed on all short channels with SCI $> = 0.15$ for each session of data per participant. Fig 3B shows the number of short channels (yellow) included in PCA. The first two principal components (PCs) that contributed the most to the short-channel responses were assumed to be the 'global' components across channels and needed to be removed. The mean ± SD contributions of the first two PCs extracted from HbO and HbR data to the short-channel responses explained 71.88% ± 7.65% and 58.66% ± 6.60%, respectively, of the variance in the signals across participants for three sessions of data collection. These two PCs were used as two regressors for a GLM with another four stimulus-specific (event-related) regressors, each for one hearing condition. Each stimulus-specific regressor was built by a box-car function for the stimulation convolved with two-gamma hemodynamic response functions (HRF), adopted from Kamran et al. [52]. Though previous studies suggested that including a derivative of the HRF would result in a more accurate estimation of the neuronal activity by accounting for shifts of time-to-peak and response window [24,53], concerns about over-parameterizing the GLM led to exclusions of derivatives in the current study. In a second method (GLM-PCA2), only the two PCs were used as regressors for the GLM, without stimulus-specific regressors. The

GLM-PCA and GLM-PCA2 methods were performed on the HbO and HbR data from each of the 30-mm channels, separately, and the product of the two PCs and the corresponding coefficients from GLM was then subtracted from each channel. The same band-pass filter and block-average were then conducted on the signals in the 30-mm channels, based on which responses in ROIs were calculated.

9. **Calculate the CNR values**. To evaluate the signal quality, the CNR values of ΔHbO responses were calculated for individual channels, by first identifying the peak of block-averaged ΔHbO responses within 5–16 s after stimulus onset. The CNR was calculated as the mean ΔHbO within 10 s centered at the peak ($HbO_{sig}$), divided by the root-mean-square of the sum of the variance of $HbO_{sig}$ and the variance of 5s baseline noise ($HbO_{noise}$), i.e., $CNR = mean(HbO_{sig})/sqrt(std(HbO_{sig})^{^2} + std(HbO_{noise})^{^2})$. The mean CNR across channels clustered in each ROI was then calculated. We expected greater than zero CNRs in all ROIs, which covered the ventral pathways of the auditory speech processing [45,46]. When individuals were listening to and processing sentences, neuronal activity from cerebral tissues in the ventral speech pathway would increase, which was associated with increased ΔHbO responses through neurovascular coupling [54].

## Statistical analysis

To evaluate the efficacy of Anti-Corr and two GLM-PCA methods in reducing the systemic noise in fNIRS signals, versus the results with no correction, a multilevel linear model was conducted on the CNR values calculated from different methods [55], with subjects being a random factor and ROIs (N = 12) being the within-subjects factor. We did not conduct a repeated-measures analysis of variance (ANOVA) as performing F-tests requires independence (lack of sphericity) between measures, whereas fNIRS responses in the current study from the 12 ROIs were dependent. Benjamini-Hochberg post-doc tests were conducted to reveal the differences between different methods.

## Results

### fNIRS data quality in the shorter and regular channels

The data quality in the short (8-mm) channels was calculated as SCI and the mean ± SD SCI values among the three sessions of data collection for all the 8 short channels in 23 participants are shown in Fig 3A. As shown in Fig 3A, the data quality in short channels varied among participants. While some participants (1–7) showed SCI close to 1 (perfect correlation) in all the 8 channels (8 dots in each column), other participants (14–23) showed SCIs near 0 or in negative range in one or more short channels. The numbers of short channels included for PCA analysis (SCI > = 0.15) in three sessions among participants are shown in Fig 3B (left). The mean and standard error of the mean (SEM) of the exclusion ratio of the short channels was 10.69% ± 3.17%. The number of regular (30-mm) channels included for further analysis for each session are shown in Fig 3B (right). The mean ± SEM exclusion ratio of regular channels was 2.80% ± 0.97%.

### Block-average responses in the 8-mm channels

After excluding the channels of poor quality (SCI < 0.15), the group average of ΔHbO and ΔHbR of the eight short (8-mm) channels (S1 –S8) were examined. Fig 4 plots the block-averaged fNIRS responses in the eight short channels. Channels S1 –S4 were on the left hemisphere and S5 –S8 were located at the symmetric positions on the right side. Group means (solid

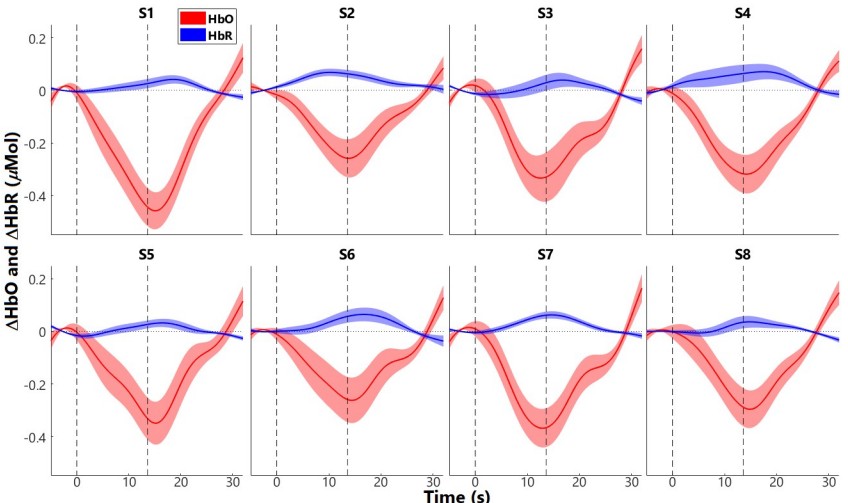

**Fig 4. Short-channel responses.** Mean (solid lines) and standard error of means (SEM; shaded areas) responses are shown for ΔHbO in red and ΔHbR in blue. The two black vertical dash lines denote the onset and offset of stimulation, which had a duration of 13.6 s. The value 0 (horizontal dash lines) was the change in response relative to baseline activity, i.e., the average of responses during the 5 s prior to the time 0. S1 –S4 are for the 4 short channels on the left hemisphere, and S5 –S8 are for the short channels in the symmetric positions in the right hemisphere.

lines) and SEMs (shaded areas) are plotted for ΔHbO (red) and ΔHbR responses (blue). Each epoch consists of 5-s baseline and 32-s responses after stimulus onset, with stimulus duration (marked by vertical lines) of 13.6 s. As shown in Fig 4, all the short channels had suppressive responses, i.e., decreased ΔHbO and increased ΔHbR after stimulus onset, opposite to expected event-evoked neuronal ΔHbO and ΔHbR—increase in ΔHbO and decrease in ΔHbR.

## fNIRS responses with and without short-channel subtraction

To reduce the systemic noise in fNIRS signals, the Anti-Corr method without short channels and the two GLM-PCA short-channel subtraction methods were applied (see Fig 2). fNIRS responses in all ROIs before and after using Anti-Corr and GLM-PCA to reduce the systemic noise were compared. Fig 5 shows the group mean and SEM ΔHbO responses without reducing the systemic components (original, red), with ΔHbO responses after applying the Anti-Corr (purple) or the GLM-PCA method (with stimulus-related and two PC regressors, yellow), or the GLM-PCA2 (with only the two PC regressors, brown). As shown in Fig 5, the original block-average ΔHbO responses (red) were generally suppressive, i.e., decrease in responses after stimulus onset and below baseline (zero). This response pattern is similar to the globally suppressive responses in the short channels (Fig 4), suggesting that the suppressive responses in the extracerebral tissue largely contributed to the signals in the regular fNIRS channels. After applying the Anti-Corr (purple) or the two GLM-PCA methods (yellow and brown), the suppressive components were reduced. The block-average ΔHbO responses across most of the 12 ROIs showed increased response after stimulus onset and gradually went back to baseline after stimulus offset, resembling the expected patterns of neuronal activity related to ΔHbO. Further, the two GLM-PCA methods showed almost identical block-averaged results, and both reduced more of the suppressive components in the fNIRS signals in all the ROIs compared to the Anti-Corr method.

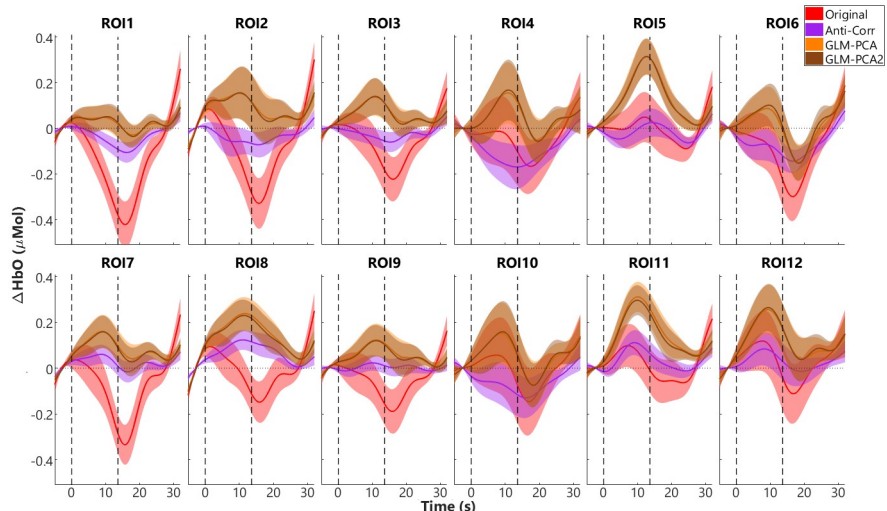

**Fig 5. Block-average ΔHbO responses in 12 regions of interest (ROIs).** The block-average results, i.e., group means (solid lines) and standard error of means (SEM, shaded areas) of ΔHbO responses without further reducing the systemic responses (Original, red) and after applying the Anti-Corr (purple), the GLM-PCA (with stimulus-related regressors and 2 PCs, yellow), and the GLM-PCA2 (with the 2 PCs as regressors, brown) methods are plotted. The two vertical black dash lines plot the onset and offset of stimuli, with a duration of 13.6 s. Responses of zero are relative to the average of 5-s baseline.

Fig 6 plots the CNR values with the Original values (red) and after applying the Anti-Corr (purple) or the GLM-PCA (yellow) or the GLM-PCA2 (brown) method for the 12 ROIs. As shown in Fig 6, applying the two GLM-PCA methods increased the CNR values compared to the Original values (red). Further, the CNRs values from GLM-PCA and GLM-PCA2 methods were almost identical. In eleven of the 12 ROIs, the CNR values from the GLM-PCA method were also greater than those from the Anti-Corr method (purple). Results from the multilevel linear model analysis showed significant differences between methods ($\chi^2(3) = 33.01$, $p < 0.001$) and between ROIs ($\chi^2(11) = 200.75$, $p < 0.001$). Results from the Benjamini-Hochberg post-hoc tests revealed significant differences in CNR values between Anti-Corr and the Original method ($p = 0.014$), between the two GLM-PCA methods and the Original ($p < 0.001$), between the two GLM-PCA and Anti-Corr ($p < 0.001$), but no differences between GLM-PCA and GLM-PCA2 ($p = 0.992$). As the choice of SCI = 0.15 was low, we also provided block-averaged results and CNR values with a cut-off of SCI = 0.75 in supplementary materials. Results from the statistical analyses demonstrated that, with SCI = 0.75 the two GLM-PCA methods significantly improved the CNR values compared to the original signals and compared to the Anti-Corr method, similar to the results with SCI = 0.15 reported above.

## Discussion

The present study compared the performance of two approaches to reducing the systemic noise in the fNIRS signals, i.e., the short-channel subtraction (GLM-PCA) method and an anti-correlation (Anti-Corr) method, which can be used when shorter channels are not available or feasible for implementation. We implemented the GLM-PCA methods with the two PCs extracted from short-channels responses, with and without stimulus-related regressors, i.e., GLM-PCA and GLM-PCA2, respectively. fNIRS data were collected from 23 young normal-hearing adults when listening to speech sentences in the left ear with bilateral noise at -10 dB StNR. fNIRS measures were examined from 12 ROIs from two hemispheres (Fig 1). To

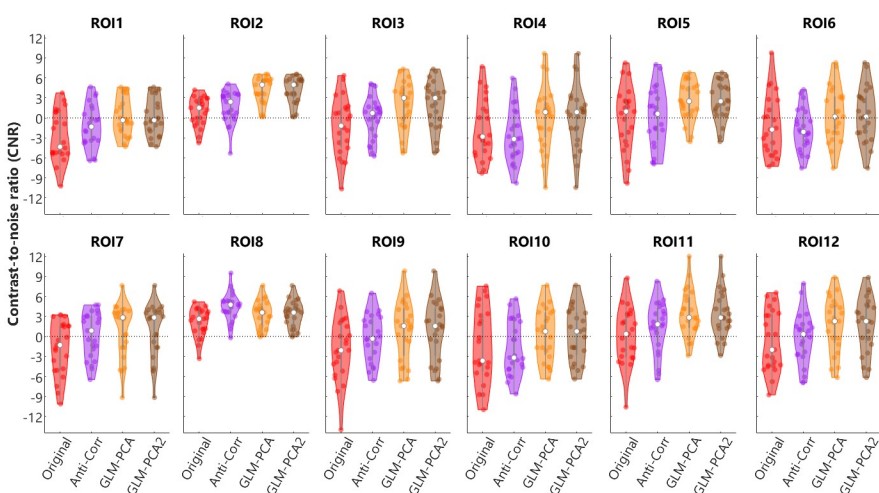

**Fig 6. Contrast-to-noise ratios (CNRs) from different methods.** Violin plots show the CNRs from HbO responses before (Original, red) and after applying Anti-Corr (purple) or GLM-PCA method (yellow), or GLM-PCA2 (brown) in individual participants in 12 regions of interest (ROIs).

assess whether any of the three methods improved fNIRS signal qualities, we calculated the CNR values from ΔHbO responses before and after applying the methods. Our results found that both the Anti-Corr and GLM-PCA methods improved the fNIRS signal qualities; there were no significant differences between the two GLM-PCA methods and they both performed better than the Anti-Corr method.

The current study used eight 8-mm channels, which were positioned on the bilateral inferior frontal and temporal lobes (Fig 1), to measure responses from the extracerebral tissues but not the deeper cerebral tissues. Our results demonstrated that responses in all the 8 short channels (Fig 4) were generally suppressive, i.e., ΔHbO responses decreased and ΔHbR increased after stimulus onset. Further, the ΔHbO responses in all 12 ROIs that consisted of regular channels showed suppressive patterns (red, in Fig 5). Similar suppressive responses were also found by Kirilina et al. [6], where a time-domain NIRS system was used with diode lasers, providing one 1-mm short channel and four 30-mm regular channels. In that study, four regular channels were positioned on the forehead with 2 on each side, and the short channel was positioned on the right forehead. Kirilina et al. [6] found that all channels showed suppressive responses when participants were performing a visual semantic categorization task. In addition to fNIRS, Kirilina et al. [6] also scanned 2 participants with fMRI and participants performed the same tasks as in fNIRS sessions. They reported that the suppressive fNIRS (ΔHbO) responses had similar patterns to the extracranial (or extracerebral) responses recorded from fMRI. Kirilina et al. [6] proposed that the suppressive responses were due to vasoconstriction in the superficial skin tissue, which might occur when participants were under mental stress or performing arithmetic tasks, resulting in a decreased blood flow, thus decreased ΔHbO responses.

Since fNIRS responses recorded from each channel were a combination of homogeneous global responses and region-specific responses from local areas [33], it is possible that the suppressive patterns found in the short channels in both our study and Kirilina et al. [6] have similar origins that are globally homogeneous, and not dependent on the channel locations or experimental tasks. In the current study, after applying the two GLM-PCA methods, the suppressive components were reduced and ΔHbO responses in most of the 12 ROIs showed

increased patterns after auditory stimulation in line with our expectations. These results suggest that the short-channel components largely contributed to fNIRS signals and that the short-channel subtraction method improved fNIRS signal qualities. However, the group average ΔHbO responses in the ROI6 and ROI10 were still suppressive after applying GLM-PCA. This result might be due to the locations of the two ROIs on the head, i.e., the temporal lobes on both sides (Fig 1C). Optodes that were involved in these regions and held by the NIRScap, were located around or above the auricles, where there was a lot of hair that could contaminate the data collection. Further, with a length of 30 mm (regular channels), near-infrared light has a limited penetration depth (about 15 mm) in adults, thus we were not able to record cerebral activity in the deeper tissues such as temporal sulcus underneath some of the channels in the two ROIs. Therefore, the fNIRS auditory-evoked signals in channels above these regions on the head might mostly consist of extracerebral components and noise, with few or no neuronal-activity related changes in hemodynamic responses.

Our results showed that the GLM-PCA (short-channel subtraction) method performed better at improving fNIRS signal qualities compared to the Anti-Corr (anti-correlation) method used previously [23–27]. This difference between methods could be contributed to by assumptions in the Anti-Corr method that may not be entirely valid. For example, the Anti-Corr method assumes that neuronal-activity-related ΔHbO and ΔHbR are anti-correlated, which is not entirely valid as ΔHbR lags behind ΔHbO. The two GLM-PCA methods using data from 8-mm channels assume that only responses from extracerebral tissue are measured from the shorter channels [38]. Further, PCA was applied to the signals from at least 4 out of 8 short channels that were spread over the bilateral inferior prefrontal and temporal cortex. The first two of the principal components were assumed to represent the systemic responses and/or residual motion artifacts that globally existed in the extracerebral tissue. Subtracting the components from the regular fNIRS channels led to greater CNR values from ΔHbO response, i.e., larger ratios of cerebral neuronal activity versus noise. Interestingly, two GLM-PCA methods, with one only using the two PCs (GLM-PCA2) and the other also including the stimulus-related regressors and (GLM-PCA), showed almost identical block averaged results and CNRs values for 12 ROIs. The results suggest that the two PCs were not correlated with the stimulus-related regressors, hence, the two GLM returned similar results for the two PCs. The finding that the two PCs in the short channels were not correlated with the event-related regressors suggests the dominance of the non-related systemic responses in the extracerebral tissue compared to the event-related systemic responses.

Despite the benefits of using GLM-PCA methods to reduce noise in fNIRS data, several limitations must be addressed. First, to retain at least 4 short channels for PCA [31], we only excluded channels with SCI less than 0.15. With this cut-off threshold, in 3 out of 23 subjects nearly 40% of short channels (Fig 3A; i.e., average of 3 to 4 out of 8 short channels) per session were excluded from data analysis. However, the criterion of SCI < 0.15 might be too low; including short channels of poor quality for PCA might introduce artifacts to fNIRS signals when performing GLM-PCA methods rather than reducing the systemic noise. Second, the ratio of short channels rejected was larger than that of the regular channels. This difference was likely due to the procedure of pushing aside the hair underneath optodes to improve the light intensity for the regular channels, but not for the short channels, since the dual-detectors were underneath the same grommets that held the light sources onto the head (see Fig 1). Third, the efficacy of the short-channel subtraction method might vary among participants and between testing sessions. In the subjects where the SCI values were above threshold but poor in most of the channels, suggesting the difficulty in recording fNIRS responses, the short-channel subtraction method might not work as well as in the subjects where fNIRS data generally had more acceptable signal qualities. Further, the data quality in short channels from some

participants also varied across sessions (Fig 3A), which could be due to the shifts of optodes on the head hence having poor contact with the skin. Though, participants were instructed to make dramatic movements to avoid the optodes shifting. In the more complicated tasks that involved a lot of head movements, it is important to better fix the optodes are on the head, to collect data of good quality. Otherwise, short-channel subtraction methods based on short channels that mostly measured motion artifacts may not be able to efficiently reduce the systemic responses in the extracerebral tissue.

## Conclusion

This study compared fNIRS signal qualities by using the current 'gold standard' method, i.e., short-channel subtraction, with an anti-correlation method that has been applied in studies when short-channels were not available or infeasible for implementation. Our results found that fNIRS responses in all the shorter channels and regular (30-mm) channels before short-channel subtraction were overall suppressive (opposite to the expected neuronal activity). In the regular channels, after the short-channel components being subtracted, fNIRS signals showed patterns similar to neuronal activity. Comparisons of results from the two approaches found that both methods improved the CNR values from ΔHbO, but the CNRs for the short-channel subtraction methods exceeded those from the anti-correlation method, indicating that short channels should be used in preference to the anti-correlation method if feasible.

## Supporting information

**S1 Fig. Block-average ΔHbO responses in 12 regions of interest (ROIs).** Results in channels with SCI> = 0.75 were included. The block-average results, i.e., group means (solid lines) and standard error of means (SEM, shaded areas) of ΔHbO responses without further reducing the systemic responses (Original, red) and after applying the Anti-Corr (purple), the GLM-PCA (with stimulus-related regressors and 2 PCs, yellow), and the GLM-PCA2 (with the 2 PCs as regressors, brown) methods are plotted. The two vertical black dash lines plot the onset and offset of stimuli, with a duration of 13.6 s. Responses of zero are relative to the average of 5-s baseline.
(TIF)

**S2 Fig. Contrast-to-noise ratios (CNRs) from different methods.** Results from channels with SCI> = 0.75 were included. Violin plots show the CNRs from HbO responses before (Original, red) and after applying Anti-Corr (purple) or GLM-PCA method (yellow), or GLM-PCA2 (brown) in individual participants in 12 regions of interest (ROIs). Results from the multilevel linear model analysis on the CNR values found a significant interaction between methods and ROIs ($\chi^2(33) = 58.74$, $p = 0.004$), and significant differences between methods ($\chi^2(3) = 239.92$, $p < 0.001$), and between ROIs ($\chi^2(11) = 89.39$, $p < 0.001$). Results from the Benjamini-Hochberg post-hoc tests revealed no significant differences in CNR values between Anti-Corr and the Original method ($p = 0.26$), but demonstrated significant differences between the two GLM-PCA methods and the Original ($p < 0.001$), between the two GLM-PCA and Anti-Corr ($p < 0.001$), but no differences between GLM-PCA and GLM-PCA2 ($p = 0.77$).
(TIF)

**S1 Dataset.**
(XLSX)

## Acknowledgments

The authors appreciate the time and support from all the research participants. We also thank various colleagues from the binaural hearing and speech lab for suggestions about experimental design and participant recruitment, including Alan Kan, Z. Ellen Peng, Emily Burg, Shelly Godar, and Tanvi Thakkar.

## Author Contributions

**Conceptualization:** Xin Zhou, Colette M. McKay, Ruth Y. Litovsky.

**Data curation:** Xin Zhou, Gabriel Sobczak.

**Formal analysis:** Xin Zhou.

**Funding acquisition:** Ruth Y. Litovsky.

**Investigation:** Xin Zhou, Gabriel Sobczak, Ruth Y. Litovsky.

**Methodology:** Xin Zhou, Gabriel Sobczak, Colette M. McKay.

**Resources:** Ruth Y. Litovsky.

**Software:** Xin Zhou.

**Supervision:** Ruth Y. Litovsky.

**Writing – original draft:** Xin Zhou.

**Writing – review & editing:** Gabriel Sobczak, Colette M. McKay, Ruth Y. Litovsky.

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
