## [Decision Letter · Decision Letter 0]

24 Jul 2020

PONE-D-20-04747

Comparing fNIRS signal qualities between approaches with and without short channels

PLOS ONE

Dear Dr. Zhou,

Thank you for submitting your manuscript to PLOS ONE. After careful consideration, we feel that it has merit but does not fully meet PLOS ONE’s publication criteria as it currently stands. Therefore, we invite you to submit a revised version of the manuscript that addresses the points raised during the review process.

The paper has been written well on an important area of fNIRS research. While praising many aspects of the article, the majority of the reviewers suggest a major revision is required to improve and deepen the introduction, method and discussion and include more details of the PCA.

Please work through and address the reviewer's comments in your revised version.

We look forward to receiving your revised manuscript.

Kind regards,

Chelsea Dobbins

Academic Editor

PLOS ONE

Journal Requirements:

"Dr. Litovsky discloses that she is a consultant for Frequency Therapeutics. The other authors have

 all certified that they have NO affiliations with or involvement in any organization or entity with

any financial interest or non-financial interest in the subject matter or materials discussed in this

manuscript."

Reviewers' comments:

Reviewer's Responses to Questions

**Comments to the Author**

1. Is the manuscript technically sound, and do the data support the conclusions?

Reviewer #1: Partly

Reviewer #2: Partly

Reviewer #3: Yes

2. Has the statistical analysis been performed appropriately and rigorously? 

Reviewer #1: Yes

Reviewer #2: I Don't Know

Reviewer #3: Yes

3. Have the authors made all data underlying the findings in their manuscript fully available?

Reviewer #1: Yes

Reviewer #2: Yes

Reviewer #3: Yes

4. Is the manuscript presented in an intelligible fashion and written in standard English?

Reviewer #1: Yes

Reviewer #2: Yes

Reviewer #3: Yes

5. Review Comments to the Author

Reviewer #1: The paper entitled “comparing fNIRS signal qualities betrween approaches with and without short channel” Is a well written paper that provides interesting information about methodological choices for the NIRS data processing. I found the paper easy to read and hypotheses and results are easy to understand. The final contribution would certainly be valuable for future functional NIRS studies. This being said I think the paper would greatly benefits from some clarifications of methodological choices and reorganization of some parts. You will find below a list of questions and recommendations related to it.

• Introduction

- A description of the different systemic components (HR, Mayer waves…) and how they interfere with the neural signal could be useful for the naive or NIRS beginners. For instance how the HR could be summed on block averaging (e.g., Kirilina 2012) and how Mayer waves’ frequency overlap with the neurovascular response.

- Why did you choose to use GLM-PCA and what's the difference with other regression like the almost conventional Ac= AL-α.AS (Scholkmann 20014 and Nirx short-channel ressources)?

- You choose PCA over ICA. Please justify this choice.

• Method

- I would suggest justifying more the limitation of the study in the comparison of only two methods. Indeed, you mentioned 5 different technics in the introduction and, if the choice of the Gold standard short channel is obvious, the choice of Anti-Corr only is not justified enough.

- I’m aware that fNIRS processing pipeline has not reached a consensus yet, however I have few concerns about your choices that you may need to clarify/justify.

- Filtering: You did not mention (or I missed it) the first band-pass filter you applied on the ΔHb [0.1 1.5] Hz in the text, hence no justification for this range is available. Since there seem to be some agreement for a [0.01 0.09] frequency band (see Pinti et al., 2019), why did you choose these values?

- Short-channel regression: You chose to apply the short-channel correction over the averaged concentration data whereas numerous researchers have stated that it is better to do it on the OD. Even the code provided by Nirx for Homer2 states that the regression should be applied on the OD. Could you justify this choice and state how this could affect the preprocessing results.

- ROIs: A figure with the ROIs would really help visualizing your results and discussion. In addition a brief justification for this choice should be written in the Method section.

- Statistics: It’s a bit annoying to discover the statistical analyses in the results section. I would prefer an a priori description of them in the Method section.

- What was the device acquisition sampling frequency?

- Participants: Please detail the material used for hearing assessments.

L120. Normal channel distance average is certainly around 30 mm but according to the 10-20 positioning system you should have variations around this distance (it can be roughly from 25 up to 40 mm according to Nirsite). You need to add more information about this issue and how it can be partially corrected using Nirx spacers (only to prevent channel from exceeding 30mm).

L122. Please provide more information about cap positioning over the head to ensure homogeneity across participants.

L133. Define IAC the first time you use it.

L169. Justify such difference compared to Pollonini. If you have clearly described why you need to have at least 4 short channels of correct quality, it seems that a SCI threshold of 0.15 is very low. Looking at your figure 3, if you choose a criterion of SCI >0.5 for instance, you'll be closer to Pollonini's and will only lose participant N° 19 while still keeping an average of 6.9 short channels for the remaining 22 participants whit a better quality.

From L189 to L211: this part could be simplified and clearer if you state only once that all three data types (Not corrected, Anti-Corr and Short-channel) where finally filtered and block-averaged in the same way.

• Results

- L353. As a reader I would be curious to have a look at the 2 principal components.

• Discussion

- As APD are known to have a better light sensitivity, I’d like to know whether your conclusion would remain the same for normal/standard photodiode.

- I’m a little bit surprised regarding the results for the auditory cortex. Since several studies have succeeded in probing this region with conventional NIRS setup (see references below), I’m not convinced about your argument. At this point, I recommend doing a Monte Carlo simulation of photon travel with your setup in AtlasViewer to discuss your result with actual depth penetration values. I’m currently doing research with a comparable montage and have quiet descent sensitivity in the superior temporal sulcus, precisely in the primary auditory cortex. Here again a picture of the ROIs would help in the results interpretation.

Refs:

Santosa, H., Hong, M. J., & Hong, K. S. (2014). Lateralization of music processing with noises in the auditory cortex: an fNIRS study. Frontiers in Behavioral Neuroscience, 8, 418.

Y. Minagawa-Kawai, K. Mori, I. Furuya, R. Hayashi, Y. Sato. Assessing cerebral representations of short and long vowel categories by NIRS. Neuroreport, 13 (2002), pp. 581-584

Plichta, M. M., Gerdes, A. B., Alpers, G. W., Harnisch, W., Brill, S., Wieser, M. J., & Fallgatter, A. J. (2011). Auditory cortex activation is modulated by emotion: a functional near-infrared spectroscopy (fNIRS) study. Neuroimage, 55(3), 1200-1207.

• Figures

Please cite image sources (NIRX/Nisite for instance)

Reviewer #2: Overall, I think that this paper still lacks strong and clear results.

The manuscript requires some additional work to better present the results and, possibly, also the data analysis.

I have two main concerns about this paper.

[1] The first is methodological.

The GLM-PCA method includes a GLM model that uses as regressors, in addition to the two principal components of the PCA on the short channels, also information about the temporal locations of the stimuli, i.e. the “stimulus-regressors”.

So the GLM is “informed” that there are the stimuli.

After, the comparison between the three pipelines is performed on the capacity of recovering the effect of the stimuli.

I think that this is not completely fair, as the results from the pipeline with short-channels come from a step that uses the same information that is “searched” to process the signals.

- What would be the result of using a GLM only based on the two main components?

- Or, conversely, what would be the results on the two pipelines not based on short-channels when a GLM with “stimulus-regressors” is used?

[2] The second pertains to the organization of the study.

This is a methodological paper with the aim of comparing signal processing pipelines.

However, there is not a strong result on the comparison here.

Instead, there are many qualitative considerations (many figures, maybe too many) but not a clear quantitative assessment.

I think this is partially due to the organization of the manuscript.

First, the Methods section completely lacks any information about how the pipelines will be compared, and on what basis the comparison methods have been defined. These methods are mentioned in the Results but without any strong rationale (see also below).

Second, the Results section presents many findings that are not strictly “results”. They are more qualitative descriptions that do not add any real contribution to the study. I think the Results should focus only on the comparison of the pipelines, which is the aim of the manuscript.

Partially, it is also due to the fact that the comparison methods seem quite arbitrary: there is not a motivation about how and why the Authors decided to use the correlation with the HRF. Is this choice supported by some functional hypothesis? Are there other papers that adopt the same methods?

On one side, I understand that it is hard to objectively compare the results of the three pipelines, having not a ground truth about what an ideal signal would look like.

However, the methods adopted rely on many - explicit or implicit - assumptions and so they appear quite arbitrary. For instance: why using the ROI? Is it correct to expect the same (temporally and spatially) brain activity response on all the ROIs? Are all ROIs involved in the processing of the stimulus? What makes the correlation with the expected HRF an appropriate quality measure?

Finally, it is partially because it is not completely clear what is the object of the evaluation.

Is it the use of short-channels, or the GLM-PCA algorithm? Or both?

If the short-channels: what about using other preprocessing algorithms?

If the algorithm: what about using the GLM-PCA on the “usual” channels?

If both: why both? And why was the GLM-PCA chosen?

- The Authors should try to better organize the manuscript and adequately motivate the methods adopted.

- The purpose of the study should be better identified and the Results section should focus only on the presentation of the quantitative results.

Then there are some major and minor issues.

Major

- The preprocessing procedure involved a step in which some channels were removed. As this step operates on each subject independently, it is important to state how many channels have been used in the downstream analysis (see Bizzego et al. 2020);

- Short channels are used to different sources of noise from the standard channel, as mentioned in the Introduction. However, in this study the Authors adopt a number of countermeasures to avoid and remove noise: they ask the subjects to remain still and they filter the signals to remove noise. The Authors should comment on that, as the adopted countermeasures could make useless the use of short-channels and other processing procedures.

- Why only the first two principal components have been used? Why two? This choice should be better motivated.

- Figure 2: it seems that the standard channels in the short-channel “track” undergo two bandpass filters: the first with 0.1 - 1.5 Hz, and the second 0.01-0.1 Hz. But this is not mentioned in the manuscript. Was the first filter really applied? If so, then the two filters would actually filter al the frequency components, as the bands do not overlap.

- Why were the channels merged into ROIs? How were these ROIs created? Do they group channels according to the Brodmann Areas? How many channels have been used for each ROI and participant?

- L. 291-293: why the multilevel linear model? What is the hypothesis that is tested? Any reference from the literature? This sentence is also unclear.

- L. 298-300: what is this follow up contrast analysis? Contrast between what? If reported then it should be described and motivated in the Methods section.

Minor

- L. 94: I think that what is compared should be the efficacy of the pipelines and not the “quality” of the signals. The quality of the signal is something that is given, the pipelines can be more or less robust to low quality signals.

- L. 119-120: “A single source …” This sentence is not clear

- L. 134-135: I think “optimization” is not an appropriate term. Optimization usually indicates a procedure where the parameters of an algorithm are tuned to maximise the efficacy.

- L. 156-162: the description of this step could be more accurate. Maybe “derivative” is more appropriate than “deviation”; I think the Authors also used the absolute values of Xi

- L. 171-173: This sentence should motivate the use of ‘4’ as the minimum number of short-channels. But it is quite unclear. What do the Authors mean by ‘low cost budget”? Why is budget important here?

The following comments are on parts of the text that I think could be removed. In case the Authors decide to put them somewhere else (e.g. Supplementary Information)

- L. 219-221: How the Authors verified this? I think this should be presented as an hypothetical explanation.

- L. 225-228: This is not a finding as the threshold was specifically set to obtain these results

- L. 237-245: the Methods section does not mention the group-averages on the short-channels; this part comes from nowhere. Why is the readed introduced to these results? What is the purpose of showing them? If reported, they should be better motivated and logically included in the procedure.

References

Bizzego et al. (2020). Commentary: Current status and issues regarding pre-processing of fNIRS neuroimaging data: an investigation of diverse signal filtering methods within a general linear model framework. Frontiers in Human Neuroscience

doi: 10.3389/fnhum.2020.00247

Reviewer #3: This paper is concerned with a comparison between two methods of removing systemic influence from the fNIRS signal. The authors have compared the anti-correlation approach with a PCA method based on short channels. This is a useful paper for researchers working with fNIRS, especially as some aspects of signal treatment and fNIRS quantification remain in development.

In this particular paper, 23 participants listened to sentences as the primary experimental task. There is no description of the sentences/stimulus material. I can see how long each lasted for and the temporal characteristics of stimulus presentation, but I would like some concrete examples of the material and confirmation that no motor response was required from the participant.

Figure 2 provided a clear presentation of the different treatments of the fNIRS data that form the basis for this paper. However, there was one aspect that I did not understand and I'm concerned that it may confound the comparison that the authors wish to make. The original data and anti-correlation data are subjected to the same signal processing as the GLM_PCA route except for a bandpass filter of 0.1-1.5Hz that was only applied to the GLM-PCA paper. What was the purpose of this filter? Was it only applied to the 8mm (short channels)? It was not clear to me from reading the paper, why this filter had been applied to that particular data set and not the other two.

The authors are very diligent with respect to filtering the signal quality utilising the SCI method developed by Pollonini et al and showing the results in Figure 3. On page 10, they state that the original method used <0.75 as an exclusion criterion, which they felt was too strict, so instead they opt for a cut-off frequency of 0.15, which is even lower. This part of the text does not make sense and should be corrected, in addition, the authors should justify the basis of their adjustment to the original exclusion criterion - not simply say that it was too strict, i.e. if they had used it, how much data would have been lost?

I would also like more details of the PCA described on page 11. My reading of the text suggests that the PCA is based on all short-channel data, or was the PCA specific to location or hemisphere? This was not clear to me. I also did not quite understand how regressors were calculated, did the PCA represent HbO or Hbb or the HRF function? Finally, the authors extracted the largest two PCs as regressors for the GLM, but I would like to know what proportion of the total variance was accounted for by these 2 PCs.

I was also very unclear about how the ROIs had been derived from the montage and the text on this topic at the bottom of page 13 should be expanded to make this clear. Perhaps the authors could indicate the ROI on the montage diagram shown in Figure 1?

When it comes to the comparison between the three data sets, I felt that the authors relied too heavily on data visualisation. The authors make the claim that GLM-PCA method reduced the suppressive component and we can see that in the plots, or at least we can for HbO, the deoxygenated signal is more difficult to see. Their method of testing correlation with HRF is OK but a little indirect. Other authors, e.g. Fairclough et al (2018) who have compared anti-correlation methods with raw data have done by comparing effect sizes in response to independent variables across different ANOVA. I would prefer the authors to adopt a similar approach by comparing activation during pre-stimulus period with activation during the presentation of stimuli via ANOVA. If the authors are concerned about sphericity, they can test each data set individually. I feel that this is important because most researchers want to know if these methods will make their data more sensitive to experimental effects - and this aspect is not really addressed in the current manuscript.

As a final point, I would like the authors to consider in the Discussion whether their results would generalise to more complex tasks that involve more head movement and a motor component. They use a task that is very controlled (no response, minimal head movement) and they should consider whether their results would generalise to different types of task.

---

## [Author Response · Author response to Decision Letter 0]

5 Sep 2020

Please see our point-to-point responses to reviewers, submitted as a separate word document (Response to Reviewers.docx), attached at the end of the submission/revision.

---

## [Decision Letter · Decision Letter 1]

10 Nov 2020

PONE-D-20-04747R1

Comparing fNIRS signal qualities between approaches with and without short channels

PLOS ONE

Dear Dr. Zhou,

Thank you for submitting your manuscript to PLOS ONE. After careful consideration, we feel that it has merit but does not fully meet PLOS ONE’s publication criteria as it currently stands. Therefore, we invite you to submit a revised version of the manuscript that addresses the points raised during the review process.

 <o:p></o:p>

The original three experts in the field have carefully reviewed the manuscript entitled, "Comparing fNIRS signal qualities between approaches with and without short channels", their comments are appended below. <o:p></o:p>

 <o:p></o:p>

All the reviewers acknowledged that the revised manuscript has been greatly improved; The first and the third reviewer satisfied the revision while the second one pointed out some concerns regarding description and presentation of methods and results.  <o:p></o:p>

 <o:p></o:p>

I will make the final judgement after receipt of your responses and the appropriate revision. <o:p></o:p>

We look forward to receiving your revised manuscript.

Kind regards,

Manabu Sakakibara, Ph.D.

Academic Editor

PLOS ONE

Additional Editor Comments (if provided):

The original three experts in the field have carefully reviewed the manuscript entitled, "Comparing fNIRS signal qualities between approaches with and without short channels", their comments are appended below.

All the reviewers acknowledged that the revised manuscript has been greatly improved; The first and the third reviewer satisfied the revision while the second one pointed out some concerns regarding description and presentation of methods and results.

I will make the final judgement after receipt of your response and the appropriate revision.

Reviewers' comments:

Reviewer's Responses to Questions

**Comments to the Author**

1. If the authors have adequately addressed your comments raised in a previous round of review and you feel that this manuscript is now acceptable for publication, you may indicate that here to bypass the “Comments to the Author” section, enter your conflict of interest statement in the “Confidential to Editor” section, and submit your "Accept" recommendation.

Reviewer #1: All comments have been addressed

Reviewer #2: (No Response)

Reviewer #3: All comments have been addressed

2. Is the manuscript technically sound, and do the data support the conclusions?

Reviewer #1: Yes

Reviewer #2: Yes

Reviewer #3: Yes

3. Has the statistical analysis been performed appropriately and rigorously? 

Reviewer #1: Yes

Reviewer #2: Yes

Reviewer #3: Yes

4. Have the authors made all data underlying the findings in their manuscript fully available?

Reviewer #1: Yes

Reviewer #2: No

Reviewer #3: Yes

5. Is the manuscript presented in an intelligible fashion and written in standard English?

Reviewer #1: Yes

Reviewer #2: No

Reviewer #3: (No Response)

6. Review Comments to the Author

Reviewer #1: The authors have addressed all my comments and recommendations. The paper gained in strength and is clearer now.

Reviewer #2: I think the analysis, results and manuscript have been improved: now the flow from the aims, methods and hypotheses and results is more logical and easier to follow.

I still have some methodological concerns:

- The authors should comment on an implicit assumption that I think it is critical: that an increase of HbO is expected during the stimulation. The CNR measure relies heavily on this assumption and would be pointless if no response is observed. The Authors should provide some references in the literature to support this assumption.

In parallel, instead of using custom ROIs, I would find more reasonable to group channels based on BAs, showing that those associated to the processing of the acoustic stimuli show greater activation.

- In any case, I urge the Authors to describe the rules they followed to compose the ROIs.

The major issue is still about the organization of the contents in the manuscript and readability of the text.

- description of methods in the introduction: I suggest to first describe all methods that do not require short channels, and then those with short channels.

- L98-104 interrupt the logical flow of the description and they raise a point that is never addressed later. Maybe this paragraph can be removed.

- L117-133 this part is very hard to follow, some sentences are also ambiguous or too complicated. I strongly suggest to rephrase and reorganize the structure of this paragraph. I understand that the qualitative description of the brain response in short channels is considered of potential interest for the reader, but it is important to give some structure to this part: try to distinguish the different aims and between qualitative and quantitative results. I would suggest to keep the term "data" only to indicate signals or experimental measures. Here the authors probably mean "information"/"description" of the functioning of the short channels.

- L53 and throughout the text: please ensure that the "Ref [x]" is an acceptable format to cite references;

- The description of the stimuli should state how many different stimuli are used, if they were repeated etc. I feel that the complete description of the experimental settings is needed to understand how the data were collected. Some of these details are reported after in the text, but I think it would be clearer to aggregate these details here. This should also be the place where the report what are the different experimental conditions and, in particular, why the chose this, and why only one.

- L152-153 The rephrased sentence is still unclear: maybe the authors just wanted to define what is considered to be a "channel" in fNIRS data? If so, also take the opportunity to clarify other points: how many total "regular" channels have been collected for each signal (should be 40: this information can be extrapolated but it is better to explicitly state that); what do they mean with "overlapping" channels: I think (hope) this does not mean that they refer to the same anatomical location (i.e.: same detector-source pair), so probably the Authors mean "near" instead of overlapping.

- I find the description of the montage (L157-168) a bit rough: what is a "bundle" in this case? please avoid ambiguous terms. Figure 1A is not very informative, it can be removed; channels in Figure 1C should be highlighted; Figure 1C and 1D can be merged if different colors are used to highlight channels belonging to different ROIs. Credits of Figure 1B are missing - I think. L166-168: I think that what authors mean is that whenever the detector was placed more than 30 mm from the source, the detector is re-positioned at 30 mm from the source, by means of a custom plastic holder.

- L182-190: as previously commented, I would avoid separating the description of the experimental design from the description of the stimuli. Also the description should be complete and authors should motivate why the only focused on one condition.

- Please use mathematical formulae and symbols; first write the equation, then describe the terms in the equation.

- L205: "0.5 - 1.5 Hz signals" does not mean anything. What do the authors mean? Are these the original signals after the 0.5-1.5 Hz bandpass filtering?

- Setting such a low SCI is quite critical. I would prefer to see the analysis on fewer subjects than a pipeline with such extreme parameter values. I would suggest to include a supplementary material with the results with the original SCI=0.75 threshold.

- The pipeline description can be improved. The authors should separately describe the steps that are applied to all methods (1-5; 6 and 9) and those that are particular of each method (7 / 8 / 9). It is hard for the reader to exactly understand what was done.

- The authors should comment on the fact that a decrease in HbO is observed in short channels. Was it expected? Why? Any similar finding in the literature.

- Figure 3B is not optimal to explain how many regular and short channels were actually used: I would try with an histogram.

Reviewer #3: I would like to thank the authors for their full and considered responses to the points raised in my review. I’m happy with their responses and have no further questions with respect to the revised manuscript.

7. PLOS authors have the option to publish the peer review history of their article (what does this mean?). If published, this will include your full peer review and any attached files.

Reviewer #1: **Yes: **Sébastien Scannella

Reviewer #2: No

Reviewer #3: **Yes: **Stephen Fairclough

---

## [Author Response · Author response to Decision Letter 1]

23 Nov 2020

Please see our point-to-point responses in the document 'Response to Reviewers.docx'.

---

## [Decision Letter · Decision Letter 2]

7 Dec 2020

Comparing fNIRS signal qualities between approaches with and without short channels

PONE-D-20-04747R2

Dear Dr. Zhou,

We’re pleased to inform you that your manuscript has been judged scientifically suitable for publication and will be formally accepted for publication once it meets all outstanding technical requirements.

Kind regards,

Manabu Sakakibara, Ph.D.

Academic Editor

PLOS ONE

Additional Editor Comments (optional):

Reviewers' comments:

Reviewer's Responses to Questions

**Comments to the Author**

1. If the authors have adequately addressed your comments raised in a previous round of review and you feel that this manuscript is now acceptable for publication, you may indicate that here to bypass the “Comments to the Author” section, enter your conflict of interest statement in the “Confidential to Editor” section, and submit your "Accept" recommendation.

Reviewer #2: (No Response)

2. Is the manuscript technically sound, and do the data support the conclusions?

Reviewer #2: Yes

3. Has the statistical analysis been performed appropriately and rigorously? 

Reviewer #2: Yes

4. Have the authors made all data underlying the findings in their manuscript fully available?

Reviewer #2: No

5. Is the manuscript presented in an intelligible fashion and written in standard English?

Reviewer #2: Yes

6. Review Comments to the Author

Reviewer #2: Dear Authors

Thanks for considering my suggestions and editing the manuscript.

I think that now the contents are more readable and understandable.

I think the question about the "suppressive" patterns in the short-channels is still open and, in my opinion, quite substantial.

The constitutive assumption behind the use of short-channels is that the signal should not be associated with the brain response.

Now, building on the interpretation you provided, based on Kirilina et al 2012, it seems that this is not really true.

I am afraid that a consequence of this is that any time you use short channels, since they present a suppressive response, the "corrected" HbO signal will present a "spurious" increased activation.

I never had any experience with short channels, and I thank you that I learned a lot from this paper.

I also understand that this issue goes beyond the scope of this paper; however, this concern remains.

I think that including a critical discussion about this issue would benefit this study (and the reader): even if it is hard to provide answers, it is better to highlight a question than leaving a black hole.

Thank you and congratulations for this study.

7. PLOS authors have the option to publish the peer review history of their article (what does this mean?). If published, this will include your full peer review and any attached files.

Reviewer #2: No

---

## [Editor Report · Acceptance letter]

10 Dec 2020

PONE-D-20-04747R2 

Comparing fNIRS signal qualities between approaches with and without short channels 

Dear Dr. Zhou:

I'm pleased to inform you that your manuscript has been deemed suitable for publication in PLOS ONE. Congratulations! Your manuscript is now with our production department. 

Kind regards, 

on behalf of

Dr. Manabu Sakakibara 

Academic Editor

PLOS ONE